# Recent Progress of Targeted G-Quadruplex-Preferred Ligands Toward Cancer Therapy

**DOI:** 10.3390/molecules24030429

**Published:** 2019-01-24

**Authors:** Sefan Asamitsu, Shunsuke Obata, Zutao Yu, Toshikazu Bando, Hiroshi Sugiyama

**Affiliations:** 1Department of Chemistry, Graduate School of Science Kyoto University, Kitashirakawa-Oiwakecho, Sakyo, Kyoto 606-8502, Japan; s.asamitsu@kuchem.kyoto-u.ac.jp (S.A.); s_obata@chemb.kuchem.kyoto-u.ac.jp (S.O.); yuzt@kuchem.kyoto-u.ac.jp (Z.Y.); bando@kuchem.kyoto-u.ac.jp (T.B.); 2Institute for Integrated Cell-Material Science (WPI-iCeMS) Kyoto University, Yoshida-Ushinomiyacho, Sakyo, Kyoto 606-8501, Japan

**Keywords:** cancer therapy, telomere, oncogenes, G-quadruplex, selective ligands

## Abstract

A G-quadruplex (G4) is a well-known nucleic acid secondary structure comprising guanine-rich sequences, and has profound implications for various pharmacological and biological events, including cancers. Therefore, ligands interacting with G4s have attracted great attention as potential anticancer therapies or in molecular probe applications. To date, a large variety of DNA/RNA G4 ligands have been developed by a number of laboratories. As protein-targeting drugs face similar situations, G-quadruplex-interacting drugs displayed low selectivity to the targeted G-quadruplex structure. This low selectivity could cause unexpected effects that are usually reasons to halt the drug development process. In this review, we address the recent research on synthetic G4 DNA-interacting ligands that allow targeting of selected G4s as an approach toward the discovery of highly effective anticancer drugs.

## 1. Introduction

A G-quadruplex (G4) is an alternative form of DNA or RNA comprising several planar layers of four guanines (G-tetrads) held together via Hoogsteen hydrogen bonding (Figure 1) [1,2,3,4,5]. The G4 DNA has long been studied from various aspects and is now considered to be an important player in biological and biomedical events, including the control of promoter activity [6,7,8,9], genome instability [10,11,12,13], benchmarks for specified chromatin remodeling and replication [14,15], and epigenetic alterations [16,17]. The sequences, 5′-G_≥3_N_1–7_G_≥3_N_1–7_G_≥3_N_1–7_G_≥3_-3′, are advocated as consensus sequences that have the ability to form intramolecular G4s [18,19], although several exceptions have been reported at the present time [20,21,22,23,24]. Extensive physical characterizations of the G4 structure by means of a series of assays based on UV and CD spectroscopy have clearly demonstrated that the structure has extremely high stability when possessing one or two nucleotide(s) between the G-tracts (*T*_m_ = ca. 70 ~ 90 °C) [25]. In parallel, a large number of defined G4 structures were elucidated at the atomic level using nuclear magnetic resonance (NMR) spectroscopy and X-ray crystallography, opening a new avenue for the rational design of G4 ligands [26,27,28,29,30,31,32,33,34]. Such high thermal stability and availability of defined structures allow us to develop applications in the field of DNA nanotechnology [35], as well as biological and biomedical studies.

In particular, given the characteristic human genome sequence of telomere (TTAGGG)_n_ that is located at the end of chromosomes and in a single-stranded context, the formation of G4 structures in this region is first assumed in the human genome [36,37]. Abundant evidence has accumulated during the past two decades that the G4 structure is truly formed in the telomere region and has an important role in telomere-end processing in cells [38]. More importantly, stabilization of telomere G4s and blockage of telomerase activities by small molecules, exemplified by telomestatin, has been shown to be a new strategy for antitumor therapy [39,40]. 

G4-forming sequences observed in the promoter of cancer-related genes have also received a great deal of attention as potential biomedical targets for antitumor therapy [8]. Quarfloxin, a ligand that interacts with G4, had completed Phase II trials as a candidate therapeutic agent against several tumors, including neuroendocrine tumors, carcinoid tumors, and lymphoma [41]. It is considered that quarfloxin disrupts the G4–nucleolin complexes of ribosomal DNA in the nucleolus, which in turn redistributes nucleolin into the nucleoplasm, where it binds to a G4 in the promoter region of the c-*MYC* proto-oncogene to inhibit its gene expression [8]. The Phase III trials for quarfloxin are currently not proceeding due to high albumin binding [42]. In addition to this report, other tumor-related genes, including hTERT [43], c-*kit* [44], KRAS [45,46], BCL2 [47], and VEGF [48], were identified as genes in which the formation of a G4 was involved in transcriptional regulation, and its stabilization by small molecules attenuated promoter activity, eventually inducing tumor apoptosis.

Relatively recent studies revealed that G4 also had an impact on differentiation- and neuron-related genes [20]. For instance, OCT4 expression may be governed, to some degree, by G4 formation at the proximal promoter in human embryonic stem cells (CCTL14) [49], whereas the excessive formation of repetitive G4 structures on an expandable (GGGGCC)_n_ in *C9orf72* gene or (CGG)_n_ in *FMR1* gene accounts for some neurogenetic disorders [50]. On the contrary, G4 can act positively in neurons, where G4 structures at the CpG island located in xl3b are recognized by ATRX, contributing to appropriate synaptic function [51].

Extensive studies of G4s and ligands that interact with them lead investigators to believe in the notion that G4s are able to widely form in guanine-rich regions of the genome [52], in the context of cellular dynamics as exemplified by transcription, duplication, and DNA repair processes, in which DNA strands are transiently dissociated to generate flexible DNA single strands. Although numerous investigators have made tremendous efforts to obtain highly active G4 ligands, and some of them have attained great success in the development of drugs in vivo [53,54,55,56]; however, these drugs are still only midway towards approval for clinical use.

One conceivable obstacle impeding the clinical application of G4-interacting molecules seems to rest with selectivity, although the global or multiple G4 targeting approaches may be effective [57,58,59,60,61], and in fact, CX-5461, a DNA G4 stabilizer, is currently at advanced phase I clinical trials for patients with BRCA1/2 deficient tumors [57,58]. Bioinformatics and next-generation sequencing (NGS) analysis estimated that 376,000 or more putative G4-forming sequences exist in the human genome [18,62]. A growing number of G4-driven genes have also been reported, suggesting the high importance of the expanded variety of G4-interacting ligands that possess differential binding profiles [8,53,54,55,56]. However, poor ligand designability arising from the topological similarity of the skeleton of diverse G4s has remained a bottleneck for obtaining specificity toward individual G4s. Very recently, investigators have entered a new phase of the development of next-generation ligands that interact with G4, in which they consider the ligand selectivity to a particular G4 to be targeted, potentially leading to the development of molecules with high antitumor activity and bioactivity with minimal antitumor therapy side effects [56]. In this review, we address the recent research progress toward developing G4-interacting molecules that exhibit antitumor activities by affecting a particular cancer-related gene with reduced off-target effects that likely rely on a clear selectivity for targeted G4s.

## 2. Global G-Quadruplex-Selective Ligands

Since G4-interacting molecules were developed based on duplex DNA-binding molecules, investigators have initially endeavored to develop G4 ligands that have a clear selectivity for G4 structures over the duplex DNA [63,64,65]. Molecules interacting with telomere G4s, the 2,6-diamidoanthraquinone derivatives, were first found to act as telomerase inhibitors by Neidle and Hurley and their group [36]. The cationic porphyrin, TMPyP4, whose planar skeleton and cationic propensity would facilitate G4 binding, was also identified as a G4 binder [66]. Telomestatin, a macrocycle naturally occurring in *Streptomyces annulatus*, was found to exhibit telomerase-inhibiting activity by binding to telomeric G4 structures, causing telomere dysfunction, including telomere length shortening and delocalization of telomere-related proteins [39,40]. That discovery was noteworthy in that it suggested the existence of G4s in vivo. These pioneering works accelerated the development of G4-selective synthetic molecules, along with the advance in rigid methods and techniques to characterize the binding profiles of such G4 ligands [21b]. In fact, several commercially available G4 ligands, such as BRACO19 [67], pyridostatin [68], Phen-DC3 [69], L2H2-6OTD [70], and L1H1-7OTD [71], all of which have negligible binding affinities to duplex DNAs, are indispensable to biochemical, biophysical, and chemical biology studies of G4s. 

## 3. Cancer-Related G-Quadruplexes and Their Interacting Molecules

### 3.1. Telomere

A telomere is a structure of the ends of a chromosome, in which a repeated microsatellite sequence and its various related proteins (called a shelterin complex), including POT1 and TRF2, which are necessary for telomere maintenance, protect the telomeric DNA from DNA repair mechanisms, and also regulates telomerase activity in mammals [72]. The human telomere region comprises a single microsatellite repeat sequence, (GGGTTA)_n_, with a 3′ overhang at its terminus (200 ± 75 nucleotides). The length of the telomere sequence gradually shortens along with DNA replication, which limits cell growth and proliferation. The G4 structures observed in this region initially received great attention because a single-stranded background offers a greater likelihood of G4 formation. Interestingly, ATRX, POT1, TPP1, and other G4-binding proteins function in the telomere processing-like recruitment of a specific protein in a G4-involved manner, whereas varied G4 helicases, including Pif1 and BLM, act to unwind the telomere G4 structures to maintain the shelterin complex [54,73]. Furthermore, because telomerase, a protein complex for elongating the telomere sequences, is highly expressed in many tumor cells, and the abnormal processing of the telomere 3′ overhang region may confer cellular immortality, a telomere G4 could be thought of as a potential biomedical target for small molecules that inhibit telomerase activity in telomerase-positive tumor cells [36,37,39,40,74].

The 2,6-diamidoanthraquinone derivatives and Telomestatin were found to be telomerase inhibitors through their binding to telomere G4s at an early stage of G4-interacting ligand studies [36,39,40]. RHPS4 was also shown to induce telomere dysfunction by disturbing the integrity of the shelterin complex in mammalian cancer cells [38]. Later relevant studies found that a large variety of alternative higher-order structures derived from the canonical telomere G4 might be adopted at the 3′ overhang region [20,21,75,76]. Those structures and their specific motifs are amenable to a gain of specificity for telomere G4s [56] (Figure 2a).

Dimeric G4 ligands target dimeric G4s. Tandemly aligned G4 ligands permit the favorable discrimination of a dimeric G4 from one that is monomeric. The dinickel salophen dimer [77], berberine dimer [78], and telomestatin derivative tetramer [79] are examples of such ligands that discriminate G4s successfully (Figure 2b). Binding to the side faces of dimeric G4s confers a preference for multimeric G4s over one that is monomeric, as exemplified by m-TMPipEOPP (Figure 2b) [80]. By contrast, chiral helical supramolecules, Ni-M, exhibit a binding preference for dimers over monomers, with a 200-fold selectivity, probably because two consecutive G4s offer a preferred binding site (Figure 2b) [81]. Conversely, another enantiomer, Ni-P, is capable of converting a monomeric antiparallel form to a monomeric hybrid form [82]. It is noteworthy that these two enantiomers display different abilities to affect telomere dysfunction in vivo; NiP, rather than NiM, preferentially inhibits cell growth in breast CSCs compared with bulk cancer cells. This inhibition relies on telomere uncapping with the delocalization of TRF2 and POT1 from telomeres, telomere DNA damage, and degradation of the 3′-overhang by G4-mediated binding of NiP [83].

The junction pocket between two G4 units also serves as a target for specific recognition. In early studies of selective G4-interacting ligands, our group proposed the possibility that a chiral helical molecule, helicene M1, allowed for the targeting of the junction cavity between two G4 units (Figure 2b) [84]. A triaryl-substituted imidazole molecule, IZNP1, was shown to be correctly positioned into the junction by molecular modeling and to exhibit a reduced binding affinity for TERRA multimeric RNA G4s (Figure 2b) [85]. Notably, this structure specificity was biologically validated, where IZNP1 caused telomeric DNA damage and telomere dysfunction without affecting several well-studied oncogenes that have monomeric G4s in their promoter regions.

Most recently, Yang and colleagues reported that a berberine derivative, EPI, discriminates a hybrid type 2 telomere G4 structure formed by a wtTel26 sequence 5′-(TTAGGG)_4_TT-3′ from the other adoptable topologies and promoter G4s (c-*MYC*, BCL2, and PDGFR), as elucidated by an NMR structural analysis (Figure 2c) [86]. In addition, EPI has the ability to convert the other conformations of the telomere G4 hybrid type 1, and antiparallel (basket type) G4s into the type 2 hybrid. Such an unprecedented propensity and selectivity relies on the stacking onto the terminal G-quartet hydrogen bonding formation with the flanking base, and the creation of a binding pocket by adaptively forming a TTA triad above the ligand. In this study, the biological impact of EPI on telomere function, including telomerase inhibition and delocalization of telomere-related proteins in cells, was not examined. However, it is conceived as that a 5′-flanking element observed in the genome context would likely serve as an adaptive and selective binding site of EPI to the telomere region, resulting in highly disruptive activity to telomere function with minimal side effects.

A long-loop DNA sequence arranged by a monomeric G4 was also amenable to molecular recognition by hybridization of the complementary strand, which was demonstrated by an atomic-level NMR analysis [20]. In particular, hybrid molecules constructed with a complementary sequence and a G4-interacting ligand would in future be added as candidates for selective recognition of telomere G4s. 

### 3.2. c-MYC

c-*MYC* encodes a multifunctional transcription factor that can act as a transcription activator of some genes involved in cell proliferation, while acting as a transcription repressor of other genes involved in growth arrest [87,88]. There are a broad variety of c-*MYC*-responsive genes that engage in important cellular functions in concert, such as cell proliferation, metabolic transformation, and metastatic capacity [89]. In tumor cells, c-*MYC* protein function is almost always activated primarily through upstream oncogenic pathways. As the overexpression of the c-*MYC* is served in various human malignancies (particularly in 80% of solid tumors), downregulation of the gene is an effective approach to cancer therapy [87,88]. However, the c-*MYC* protein is generally considered to be an undruggable target by small molecules owing to its short half-life, large dimension, and unstructured nature [8].

The c-*MYC* promoter region contains the nuclease hypersensitive element (NHE) III_1_, which is located −142 to −115 base pairs upstream of the P1 promoter (Figure 3a). There is one putative G4-forming sequence (PQS) in this element, which is capable of forming a nonduplex species, possibly accompanied by local unwinding or melting of the duplex structure under the influence of negative supercoiling stress [90,91,92]. Structural dynamics in this region have also been considered to be a possible key mechanism in certain carcinomas, largely to govern c-*MYC* transcription, and the formation of a G4 is likely to act as a downregulator. Hence, G4-interacting ligands may contribute to suppression of downstream c-*MYC* expression by ligand-mediated G4 stabilization [7]. In this context, c-*MYC* targeting G4-interacting ligands have been studied during the past two decades with an aim toward drug applications for antitumor therapy.

An exon-specific RT-qPCR assay using two pairs of Burkitt’s lymphoma cell lines (CA46 and RAJI), devised by Brooks group, allowed us to evaluate whether G4 ligands directly act for c-*MYC* G4 or not in cells [93,94]. This assay relies on the unique reciprocal translocation of c-*MYC* gene locus in the two cell lines by the translocation in CA46, where exon 1 is separated from exons 2 and 3; the G4-driven transcriptional activity was only maintained for exon 1, whereas these three exons and the PQS remained in tandem after the translocation in RAJI. That is why the use of primers specific for the exons 1 and 2 in a qPCR permits the demonstration of the c-*MYC* G4-mediated control by G4 ligands in the cellular context. In fact, a G4-interacting ligand, GQC-05 (NSC338258), was shown to directly suppress gene expression in a c-*MYC* G4-mediated manner using this assay [93]. This system is now a standard method for studying the intracellular activities of the c-*MYC* G4 targeting small molecules.

Recently, G4-interacting ligands that have a clear selectivity for c-*MYC* G4 were reported, and some of these proved to have effective antitumor activities, probably by reducing off-target effects as seen in biological experiments.

Sequence-selective recognition of duplex DNAs adjacent to an intended G4 serves to gain specificity in the genomic context. Our group has developed hybrid molecules, where a cIKP as a G4-binding motif is covalently linked to a sequence-selective duplex-binding molecule, pyrrole-imidazole polyamide (PIP) [95]. These hybrid molecules display simultaneous and synergistic recognition of both the c-*MYC* G4 and its proximal duplex (Figure 3b). Among these, hybrid 3 exhibits a 4.6-fold selectivity for a G4/duplex substrate that was predicted to be the most likely off target (*K*_D_ = 3.7 × 10^−9^ versus 17 × 10^−9^ M) [95]. The PIP part of the hybrid molecules accounts for their selectivity—PIPs are able to recognize a G/C base pair by antiparallel pairing of an imidazole/pyrrole moiety and an A/T or a T/A base pair by antiparallel pairing of a P/P pair. A β-alanine/β-alanine pair reads an A/T or a T/A base pair in a way similar to a P/P pair [96]. This ligand-design methodology is, in theory, applicable to the selective targeting of a broad variety of designated G4s in the genome.

Dash and colleagues reported that a crescent-shaped thiazole peptide, TH3, preferentially stabilized c-*MYC* G4 over G4s in other promoters (c-*kit* 1, c-*kit* 2, and *BCL2*) and duplex DNA (Figure 3b) [97]. The binding preference was confirmed by *K*_D_ values calculated using fluorimetric titration assays, in which TH3 showed four- to fivefold higher binding affinity to c-*MYC* G4 (*K*_D_ ~ 0.3 μM). Importantly, the preference in vitro was confirmed biologically. Western blotting, qRT-PCR, and luciferase reporter assays using cancer cell lines (HeLa and A549) revealed that TH3 is able to repress the transcription of c-*MYC* at a protein level without affecting *BCL2* expression, suggesting the preferential recognition and stabilization of the c-*MYC* G4 by TH3. Such preferential targeting by TH3 might result in selective antiproliferative effects against the cancer cells over normal human cells (NKE). Molecular insight from an NMR analysis suggested that TH3 interacted with an AT-rich capping structure at both 5′ and 3′ ends that was unique to the c-*MYC* G4 structure.

Tan and colleagues developed a new four-leaf-clover-like molecule, IZCZ-3, that preferentially binds to, and stabilizes, c-*MYC* G4 (Figure 3b) [98]. This compound is a conjugate of diaryl-substituted imidazole with a carbazole moiety. Previously, a triaryl-substituted imidazole unit was shown to be a G4-selective ligand, and its derivatives were shown to be distinct probes that were able to discern different G4s, while the carbazole moiety and its derivatives showed significant binding affinities toward the c-*MYC* G4 [98,99,100,101]. By combining these features, these investigators expected that the conjugation of a triaryl-substituted imidazole scaffold with a carbazole moiety might be a suitable way to create c-*MYC* G4 ligands with high affinity and selectivity. Among the conjugates designed, IZCZ-3 was found to have about eightfold preference for the c-*MYC* G4 (*K*_D_ ~ 0.1 μM) over the G4s in the promoters for other genes (*VEGF*, *BCL2*, c-*kit*1, and *KRAS*), as demonstrated by fluorescence titration experiments. They further confirmed its biological activities extensively using western blotting, exon-specific RT-qPCR [93,94], and a luciferase reporter assay, demonstrating that IZCZ-3 was able to repress the expression of c-*MYC* at the protein level in a c-*MYC* G4-mediated manner. More importantly, this ligand showed cytotoxicity against cancer cell lines overexpressing c-*MYC* but not against human normal cells or primary mouse cells, suggesting reduced side effects based on the G4 selectivity. The investigators confirmed that the cytotoxicity induced by IZCZ-3 likely originated from the downregulation of c-*MYC* expression by checking the expression of certain cell cycles and apoptosis regulators associated with c-*MYC*. Moreover, the antitumor activity of this molecule was demonstrated in vivo using xenograft mouse models, showing an inhibition of the tumor growth similar to an anticancer drug, doxorubicin. Collectively, IZCZ-3 showed a high affinity and discriminating capability for c-*MYC* distinct from other parallel G4s, as well as antiparallel or hybrid G4s, whose differential binding profile conferred a selective inhibitory ability against c-*MYC*-driven tumors.

A new approach, using a small molecule microarray screen of 20,000 compounds using fluorescently labeled c-*MYC* G4 DNA, successfully identified a c-*MYC* G4-selective ligand (Figure 3b) [102]. The direct interaction between c-*MYC* G4 and its G4-meditated influence over DNA replication was validated by an SPR-binding assay and a PCR stop assay. Importantly, the ligand inhibited c-*MYC* transcription and reduced cell viability in a panel of myeloma cell lines, whereas it exhibited minimal effects for a cell that harbors a c-*MYC* translocation, deleting the G4-forming element in the promoter and normal blood mononucleocytes. Furthermore, gene expression analysis and RT-qPCR demonstrated that the ligand did not change other G4-driven gene expression, including genes for BCL2, KRAS, HIFA, VEGF, and, Rb1, indicative of excellent selectivity. Similarly, another screening approach permitted the selection of a c-*MYC*-G4-interacting ligand [103]. Although the detailed selectivity to c-*MYC* G4 over the others was not mentioned in the report, the identified compound (Tz 1) allowed for an excellent repression of c-*MYC* at low micromolar concentrations in cultures of colorectal carcinoma (HCT116) cells, and the G4-mediated action was biologically validated by an exon-specific RT-qPCR assay [93,94] in Burkitt’s lymphoma (CA46) cells (Figure 3b).

### 3.3. VEGF

Tumor progression and metastasis render the tumors more mature and malignant than undeveloped neoplasms, eventually resulting in deterioration and immortality. Overexpressed VEGF (Vascular Endothelial Growth Factor) proteins, including VEGFA, VEGFB, VEGFC, VEGFD, VEGFE, and PIGF in tumor cells, are responsible for induced neovascularization. The expression of human VEGF, which is frequently elevated in many types of cancers, is regulated mainly at the transcriptional level [104,105]. In a reporter assay system using several cancer cell lines, regulation of VEGF was regulated basically by a sequence from –85 to –50 relative to the transcription start site containing five arrays of more than three consecutive G-tracts, which are likely to adopt the G4 form of DNAs [106,107]. VEGF is an attractive target molecule for malignant tumor therapy, and antibody drugs targeting it have been approved for solid tumor treatment [108,109]. Interestingly, *VEGF* has a promoter region, in which the G4-forming sequences are located [48]. The sequences are also consensus sequences for transcription factors, such as Egr-1 and Sp1, suggesting that the dynamic equilibrium of DNA forms in this region regulate VEGF expression (Figure 4a) [106,107]. In addition, hormone response element (HRE, 5′-ATACGTG-3′), situated 969~975 nucleotides upstream of the transcription start site, also regulate the transcription of *VEGF* gene (Figure 4a), and pyrrole-imidazole polyamides (PIPs) targeted to this element were shown to repress the VEGF gene expression by interfering with the binding of HRE-binding transcription factor (HIF-α) [110]. 

Initially, the interaction of TMPyP4 and telomestatin with G4 oligonucleotides was proven to unwind the duplex DNA oligomer into an ssDNA oligomer and to stabilize the G4 structure [48], and Se2SAP, a global G4-interacting ligand, efficiently suppressed VEGF expression in two adenocarcinoma cell lines (HEC1A and MDA-MB-231) [111]. These data offer the possibility that the transcription regulation of VEGF is controllable by ligand-mediated G4 stabilization and led to the application of G4-interacting ligands for cancer therapy. Similarly, a perylene monoimide derivative, PM2, was found to be a VEGF downregulator, likely by direct interaction with the G4 structure [112] (Figure 4b). A quindoline derivative, SYUIQ-FM05, also demonstrated strong interactions with a VEGF G4 and exhibited potential antiangiogenic and antitumor activities [113] (Figure 4b). On the basis of these successful studies, several VEGF G4-preferred ligands have been developed through low-volume screening by means of docking and spectroscopic approaches [114,115]. Biological activities of these ligands have not been examined thus far, and therefore we await a future study for their determination.

### 3.4. BCL2

BCL2 gene is recognized as an apoptosis-related gene whose translated product resides on the cytoplasmic face of the mitochondrial outer membrane and acts to suppress the movability of apoptosis-induced proteins by controlling mitochondrial membrane permeability [116]. Overexpressed BCL2 is associated with aberrant carcinoma growth in various human diseases, particularly with solid tumors, such as lymphomas, non-small-cell lung cancer, myeloma, and melanoma, having been recognized as targets for cancer therapy in the past three decades [117]. Several approaches have been made to downregulate the BCL2 expression in cancer cells toward cancer therapy, including using small molecules to disrupt protein–protein interactions [118], antisense oligonucleotides [119], and peptidomimetics [120]. Overexpression of BCL2 is also indicated to be a principal element of chemoresistance, particularly for lymphocytic cancers [121,122]. For instance, transfection of BCL2 into A549 cells induced a resistance to the apoptotic effect triggered by triazine derivative 12459, a G4-interacting ligand that inhibits telomerase activity [123]. As another approach, the molecular decay effect by the guanine-rich AS1411 aptamer that can be stably folded into a G4 structure causes the destabilization of BCL2 mRNA and degradation with RNase by interfering with the binding of nucleolin to the AU-rich element of BCL2 mRNA, eventually inducing apoptosis [124]. This approach is reminiscent of the involvement of G4 formation in gene expression.

Amplification and translocation of BCL2 genes are shown to be equally common mechanisms that cause its overexpression in human cancer cells [125]. The human gene for BCL2 includes P1 and P2 promoters and has multiple transcription start sites (Figure 5a). The major transcription regulation is less driven by a TATA-box in promoter 2, while the P1 promoter that is situated 1386–1423 nucleotides upstream of the translation start site has been largely implicated in the control of BCL2 transcription [126,127]. The GC-rich element exists 1490–1451 nucleotides upstream of the P1 promoter, where multiple transcription factors have been said to be implicated in BCL2 gene expression, including Sp1 [126], WT1 [128], E2F [129], and NGF [130]. The multiple G4 structures in this region were well elucidated by Hurley and Yang [47,131,132]. The regulatory effect of the G4s was suggested by luciferase reporter assays, in which mutation or deletion of this region resulted in an increase in promoter activity in B lymphocytes (DHL-4) [127] or human promyelocytic leukemia (HL-60) cells [133]. More recently, Onel, Yang, and coworkers demonstrated by a luciferase reporter assay using BCL2 promoter and mutated sequences that the formation of another G4 situated almost on the upper region of the P1 promoter attenuated the promoter activity (Figure 5a) [134]. Based on these studies, an approach to stabilizing the G4s formed in the regulatory element and attenuating the promoter activity by ligands has also been studied for cancer therapy, similar to the small molecule targeting of c-*MYC* G4.

Proof-of-concept studies were performed by Huang, Gu, and colleagues, in which SYUIQ-FM05, as mentioned in the last section, was able to repress BCL2 transcription with a negligible influence on the case, using a promoter mutated to abolish its ability to form G4 in a reporter assay [133]. Moreover, these investigators demonstrated that the ligands induced apoptosis of HL-60 cells. Very recently, furo[2,3-*d*]pyridazin-4(5*H*)-one derivatives were screened as a new class of G4-interacting ligands for BCL2-targeted therapeutics, and two hit compounds identified were found to bind to BCL2 G4 structures with clear preferences over c-*kit*, c-*MYC*, and telomere G4s, as well as dsDNA (Figure 5b) [135]. Importantly, one of the two compounds repressed BCL2 expression, showing a remarkable cytotoxicity to Jurkat (human acute T cell leukemia) cell lines. Additionally, we introduce a probe preferentially targeting BCL2 G4, carbazole TO, whose fluorescence intensity is more greatly enhanced in the presence of BCL2 G4 than it is in the presence of G4s of other promoters, telomere G4, ssDNA, and dsDNA (by approximately 2.6–19 fold) (Figure 5b) [136]. Such a ligand having excellent selectivity to a particular G4 may be applicable to a potent facile light-up probe for BCL2 G4 in admixture contexts, such as cellular environments in diagnostics, therapeutics, and biosensors.

In a different way, invading PNA to directly hybridize the cytosine-rich strand and to indirectly stabilize the BCL2 G4 facilitates G4 formation [137]. This would be a versatile way to target and stabilize a particular G4.

In addition to the G4s, the i-motif, another form of DNA that forms in cytosine-rich sequences, is involved in transcriptional regulation, in which the binding of hnRNP LL to the i-motif structure likely activates BCL2 gene expression [138,139]. Moreover, a molecule interacting with the i-motif, IM-48, was identified modulating BCL2 gene expression by affecting the dynamic equilibrium of the i-motif and the flexible hairpin form [138,139], opening a new avenue to modulate the expression of BCL2 more precisely. Targeting such canonical DNAs formed in the regulatory element of the promoter may be an effective way to target a particular target to combat the tumor.

### 3.5. c-Kit

The c-*kit* proto-oncogene encodes a receptor tyrosine kinase that is bridged and activated by the binding of dimerized stem cell factors (SCF), and in turn stimulates proliferation, differentiation, and survival in hemopoietic precursor cells [140,141,142]. Malfunctions of Kit acquired by overexpression or mutations have been associated with several diseases, including gastrointestinal stromal tumors (GIST), mastocytosis, and acute myelogenous leukemia (AML) [143,144,145], and although the kinase inhibitor Imatinib (Glivec) has been successfully developed as an FDA-approved drug for GIST, long-term exposure often causes secondary mutations at exons 13, 14, or 17, which encode tyrosine kinase domains [146]. Notably, drug resistance derived from mutations at exon 17 is found to severely attenuate the therapeutic effect of imatinib [147]. A compelling approach to fundamentally suppress c-*kit* expression is highly desirable.

The human c-*kit* promoter is devoid of both TATA and CCAT boxes [148,149]. Instead, the region within 200 bp upstream from TSS is highly rich in GC content, where several transcription factors, including MAZ in human normal fibroblasts and SP1 in hematopoietic cells and carcinomas, are implicated (Figure 6a) [150,151]. Two well-defined G4 structures were resolved [44,152], and the three-dimensional structural dynamics are shown to be involved in the regulation of c-*kit* gene transcription, accelerating the development of c-*kit* G4-preferred ligands [153,154,155,156]. The modulation of such structural dynamics by small molecules is effective for suppressing gene expression and exhibits an apoptotic effect.

The pioneering work performed by Balasubramanian and colleagues warrants attention, where low-volume screening was conducted using six isoalloxazine ligands. Among the ligands, 1a and 1d, possessing three *N*,*N*-dimethyl amine-substituted and one *N*,*N*-dimethyl amine/2 fluorine-substituted tails, respectively, show a clear binding preference to c-*kit* G4s over a telomere G4, as demonstrated by an SPR-binding assay, and an inhibitory effect for the gene expression in two different cancer cell lines by RT-qPCR assays (Figure 6b) [157].

The feasibility of targeting the c-*kit* promoter G4 by small molecules was further confirmed in patient-derived GIST cells. Neidle and colleagues demonstrated that a naphthalene diimide derivative strongly stabilized c-*kit* G4 structures and almost completely reduced the level of protein expression, resulting in a strong effect on growth arrest in the GIST tumor cells (Figure 6b) [158].

To obtain a more bioactive c-*kit* G4-interacting ligand, several cell-based screenings of G4 ligands have been performed. This approach is expected to overcome the incompatibility between the outcomes of in vitro and cellular applications, which is often seen in ligand discovery based in vitro. To give examples, luciferase reporter assays performed on a 96-well plate using the human gastric carcinoma cell line (HGC-27) led to the discovery of two benzo[*a*]phenoxazine (BPO) derivatives as potent c-*kit* G4 ligands (Figure 6b) [159]. Subsequent RT–qPCR and SPR-binding analyses confirmed that these two molecules acted as endogenous c-*kit* suppressors in an HGC-27 cell line, probably through binding to c-*kit* promoter G4s. Similarly, two quinazolone derivatives were identified that could downregulate c-*kit* expression at the protein level (Figure 6b) [160].

### 3.6. Human Telomerase Reverse Transcriptase

Human telomerase reverse transcriptase (hTERT), which encodes the catalytic subunit of telomerase, has commanded considerable attention as a compelling biomedical target, particularly for cancer, as elevated TERT expression is observed in ~90% of human cancer cells, whereas it is normally silenced in most normal cells [161,162]. Aberrantly expressed TERT accelerates telomerase activity irregularly to maintain the telomere length [163]. Other than its canonical role in maintaining telomere length, TERT suppresses BCL2-dependent apoptosis [164] to regulate the chromatin state [165] and DNA damage responses [165,166], and to promote c-*MYC* and Wnt-driven cellular proliferation [167,168].

The mutations that were identified in >70% of melanomas partially account for the elevated level of TERT expression [169]. C to T mutations in the sense strand (G to A mutations in the antisense strand) in the TERT promoter highly activate transcription by creating a new consensus sequence for the binding of ETS/TCF (E-26/ternary complex factor) [170]. Patients who have tumors expressing elevated levels of TERT exhibit even worse survival rates than those who have tumors expressing relatively lower levels of TERT [171]. These observations clearly indicate that TERT promoter targeting based on the mutations might have a great impact on tumor therapeutics covering a wide range of tumors (Figure 7a).

We would like to highlight a unique approach to addressing the issue of hTERT downregulation on the basis of the mutations in a G4-mediated manner. The region located approximately -90 to -22 upstream of the transcription start site was abundant in GC base pairs and has multiple PQSs (Figure 7a) [43]. In particular, tandemly aligned G4s formed by the entire PQSs have been suggested to be a key mechanism in maintaining the normal transcriptional levels of the hTERT gene [43,172]. Interestingly, the involvement of the dynamic equilibrium of such three-dimensional structures of DNA upon the somatic mutations (G/C to T/A) in this region is attributable to activated TERT expression [172]. Hurley and colleagues developed a small molecule that binds to the higher-order G4 structure observed in the hTERT promoter via dual-motif targeting for the G4 and the mismatched duplex stem loop (GTC365, Figure 7b) [172]. The mutations situated at a mismatched duplex-stem loop within G4 structures alter the folding pattern of G4s to increase the gene expression, likely by inhibiting a certain transcription repressor. The interplay of GCT365 and the element was proposed to act like a chaperone to steer the correct folding seen in the wild-type G-rich element of the promoter, resulting in the reactivation of the silencing function.

### 3.7. KRAS

The RAS gene family, including *HRAS*, *NRAS*, and *KRAS*, was first discovered in human tumors as driver oncogenes and has long been recognized as an important therapeutic target. Mutation of *KRAS* is one of the most oncogenic driver mutations in pancreatic, colorectal, and lung cancers, and plays a role in acquiring and increasing the drug resistance [173,174]. Hence, direct targeting of active *KRAS* by small molecules was considered to be a compelling strategy for combating *KRAS* mutant tumors, yet it remains at an unsuccessful stage. Recently, our group has developed a novel approach that directly targets the mutant DNA using an alkylating pyrrole-imidazole polyamide (PIP) molecule, which is capable of selectively alkylating oncogenic codon 12 mutant DNA and causing strand cleavage, and consequent tumor growth suppression in a tumor xenograft model of cancer in mice [175].

G4-mediated promoter targeting is also reported. The NHE in the *KRAS* proximal promoter is highly abundant in G-rich sequences, and several transcription factors interact with a G4 structure formed in this region (Figure 8a) [176,177,178]. A polypurine G-rich element, located approximately –300 to –100 nucleotides upstream of the exon 0/intron 1 boundary in a murine or human genome, is likely to be a component of the promoter activity, and includes multiple PQSs [45,46,176,177,178,179,180]. Importantly, pyrene-modified oligonucleotides that were devised to be a more stable form of the *KRAS* G4 formed in the PQS1 were able to attract the factors essential for transcription and to exhibit a strong antiproliferative activity through a G4-decoy effect in pancreatic cancer cells [181].

Small molecule targeting for the *KRAS* G4 would also be a promising strategy for suppressing gene expression. Indeed, Paulo and colleges have prepared a small library of indolo[3,2-*c*]quinolines (IQc), and found that two triple-cation derivatives had a clear preference for *KRAS* G4 over a telomere G4 and duplex DNA, and acted as superior downregulators of *KRAS* expression and inhibited mutated KRAS expression in HCT116 and SW620 cells [182]. The data presented suggest that direct targeting of the *KRAS* G4 at the transcription level in *KRAS* mutant tumors by small molecules selective for G4 might be a promising therapeutic strategy for tumors. More recently, comprehensive studies on the human KRAS promoter, in terms of the possible G4 structures and their relevance to the promoter activity, which were performed by Brooks group, revealed that a newly discovered G4 formed in the PQS2 more critically affected the attenuation of the promoter (Figure 8a) [46]. These findings would contribute to the creation of more efficient and selective G4-mediated transcriptional repression by ligands in a future study.

### 3.8. c-Myb

c-*Myb* is largely expressed in an early stage of the differentiation of hematopoietic cells, and its expression is gradually decreased toward the end of their differentiation [183]. It encodes a transcription factor that plays a critical role in the proliferation, differentiation, and survival of hematopoietic progenitor cells; c-*myb* was identified by the discovery of v-*myb* found in avian myeloblastosis virus and E26 [184]. This gene is also recognized as a proto-oncogene, high expression of which is related to promoting the development of hematological cancers and adenocarcinomas [185,186,187,188,189] by a mechanism based on its canonical proliferative property.

The regulation of c-*myb* expression at a transcription level relies on multiple activating and repressing transcription factors in a cell-type-dependent fashion [190,191,192,193,194,195]. Notably, a region in the promoter with three (GGA)_4_ triplet repeats beginning 17 bp downstream of the transcription start site on the antisense strand is implicated in the promoter activity through its formation of thermally-stable higher-order parallel G4 structures [196,197,198]. Partial deletion of the (GGA)_4_ triplet repeats blocks the ability to form the dimerized G4 which enhances the promoter activity, suggesting that the G4 structure forms by utilizing the three (GGA)_4_ triplet repeats, which function as a negative regulator of the c-*myb* promoter activity [196]. Additionally, MAZ protein may bind to the c-*myb* G4 structure and negatively regulate the promoter activity.

The GC-rich promoter region in the DNase hypersensitive elements (to approximately 850 bp upstream of the transcription start site) is also reminiscent of the putative G4 formation and its involvement of transcription regulation [199,200]. Recently, the group led by Yuan used a reporter assay system to extensively examine the way that folded and unfolded G4s affect c-*myb* expression through their effect on c-*myb* promoter activity [201]. In this system, four PQSs in the c-*myb* promoter were selected as potential G4 formation elements, and the activity in MCF-7 cells was compared with that in cells with a promoter-containing plasmid, where mutations were made so as not to form G4s in the respective PQSs (Figure 9a). The promoter activity of the PQS1-mutated plasmid was markedly reduced, whereas the PQS1, −2, and −3-mutated plasmids exhibited no significant changes in these promoter activities. These data strongly implied that the transcription regulation on those G-rich sequences was considerably mediated by the formation of the G4 structures on the PQS1 element in MCF-7 cells, where the binding of a transcription suppresser was likely impeded. The newly discovered c-*myb* G4-interacting ligand, topotecan, increased the transcription level in the wild-type plasmid without affecting the use of the PQS1-mutated plasmid (Figure 9b). This downregulation effect was confirmed in the endogenous c-*myb* expression at the protein level. Since the binding specificity of topotecan to the c-*myb* G4s among the other G4s has never been mentioned, further studies are needed.

When the focus moves more specifically to diseases, c-*myb* is identified as a target in glioma stem cells for glioblastoma multiforme (GBM) therapy, in which expression was considerably elevated in GBM tissues relative to normal tissues [202]. Interestingly, telomestatin, a ligand interacting with G4, globally impairs the maintenance of the GSC stem cell state through an apoptotic pathway, largely by reducing a c-*myb* expression in vitro and in vivo. Although the direct interplay of telomestatin and c-*myb* G4s in the promoter has not been examined, these observations offer the possibility that direct targeting of c-*myb* G4 DNA is a compelling therapeutic approach to GBM treatment.

### 3.9. Others (PDGFR-β, PDGF-A, STAT3, FGFR2)

Other G4s formed in putative regulatory elements in the promoters of cancer-related genes have been reported, and are proposed as targetable by ligands interacting with G4 (in promotors in genes for PDGFR-β [203], PDGF-A [204], STAT3 [205], FGFR2 [206], etc.). For instance, GSA11129, which can interact with a G4 in the promoter of the gene for PDGFR-β to shift the equilibrium to a G4 species, was demonstrated to reduce the transcription level and to inhibit PDGF-β-driven cell proliferation and migration [203]. The G-rich element of the proximal promoter in the gene for PDGF-A also forms a stable G4 structure, even in the duplex context, and TMPyP4 reduced the basal promoter activity of *PDGF-A*, suggesting that targeting the *PDGF-A* G4 by the ligand specific for this G4 may be feasible as a cancer therapy for gliomas, sarcomas, and astrocytomas [204,207,208,209,210,211].

## 4. Alternative Nucleic Acid Form as a Biomedical Target—G-triplex

The G-triplex was initially regarded as a transient DNA form and a possible intermediate in the G4 folding process [212]. A growing body of literature suggests that such a structure forms stably under physiological conditions [75,213,214,215,216]. Along with their potential biological significance, small molecules targeting G-triplexes increasingly command considerable attention. Acridone–PNA conjugates highlight dual-site targeting by a planar acridone moiety appended to a Gly-GGG-Lys PNA sequence (Figure 10a) [217]. The PNA moiety associates with one guanine of three G-tetrads, to form a hybrid PNA+DNA G4. This ligand is thought to prefer a G-rich sequence in a single-stranded context over a pre-folded G4; thus, it might be especially useful for targeting G-triplex structures in such cellular dynamics. A dihydropyrimidin-4-one derivative is identified as a G-triplex and G4 ligand from the Mcule chemical database, by using simple docking programs (Figure 10b) [218].

Our group has devised a nanoplatform constructed by DNA origami for studying such intermediates of G4, such as G-triplex and G-hairpin, and has found that pyridostatin (PDC), a well-known ligand interacting with G4, unexpectedly recognized the G-triplex and G-hairpin structures (Figure 10c) [219]. Considering this, the ability to recognize the intermediates of G4 might be an essential component for the high binding affinity, selectivity, or inducing ability of the G4 structures from the stable duplex or single-stranded DNA. The platform manifests the power to assess unprecedented G4-binding properties of a ligand.

## 5. Spatially Indirect G4 Targeting by Py-Im Polyamide Molecules

By taking advantage of the sequence-selectivity of PIPs, as mentioned earlier, a head-to-head-type polyamide dimer was developed for the selective targeting of a designated G4, which displayed an inducible effect toward G4 formation (Figure 11) [220]. This strategy relies on a totally new ligand design based on the targeting of the duplex region on either side of an intended G4 in the genome. Although the results presented are still preliminary, such simultaneous dual-duplex binding across the targeted G4 has the potential advantages of being (1) independent of G4 topology, (2) targetable to other higher-order DNA structures such as i-motifs, and (3) programmable for a large variety of targets at different genome locations.

## 6. Summary and Outlook

Although a number of biomedical targets in the approach toward tumor therapy are recognized and investigators are pursuing the development of drugs, results have been limited thus far. The G-quadruplex (G4) has been recently considered to be a potential biomedical target, particularly for tumor therapy, and a considerable body of evidence has been accumulating that G4-interacting drugs exhibit good antitumor activities. As protein-targeting drugs face similar situations, G4-interacting drugs displayed low selectivity to the targeted G4 structure, mainly because of the similar skeletons among different G4 forms observed in the genome. In this review, we addressed the ligands interacting with G4 that were devised to gain selectivity for a particular G4 structure and to exhibit selective bioactivity for tumor cells. The selectivity issues remain incompletely solved, but if accomplished, would substantially impact cancer therapy.

The G4-driven oncogenes introduced here are known to be usually well-correlated and concertedly influence tumorigenesis, tumor growth, and malignant transition [164,167,168,190,221,222]. Although this relationship is not fully elucidated, combinatorial approaches may be a good option for further therapeutic advancements [223].

Importantly, G4 has an influence, not only on telomere or tumor-related genes, but also on differentiation- and neuron-related genes. As mentioned earlier, OCT4 expression may be governed to some degree by G4 formation at the proximal promoter in human embryonic stem cells (CCTL14), as shown by a reporter assay system [49], while the excessive occurrence of poly G4 structures on the expandable repeats accounts for some neurogenetic disorders [50]. On the contary, G4 structures at the CpG island located in xl3b are recognized by ATRX, contributing to appropriate synaptic function [51]. These data clearly imply an even more global impact of G4s on overall ontogenesis. In addition to the research direction of specific G4 ligands toward cancer therapy, studies on specific G4-interacting ligands may also focus on these issues and depict an entire map of G4 functions.

## Figures and Tables

**Figure 1 molecules-24-00429-f001:**
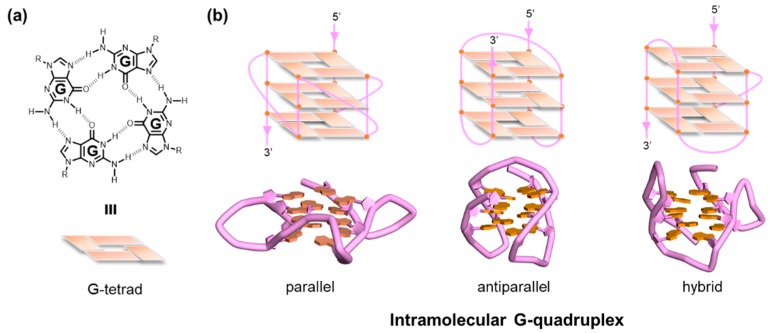
(**a**) Structure and schematic illustration of a G-tetrad. (**b**) Schematic illustrations of typical intramolecular G-quadruplex (G4) structures: (left) crystal structure of parallel type telomere G4 (PDB code: 1kf1); (center) solution structure of antiparallel type telomere G4 (PDB code: 143d); (right) solution structure of hybrid type telomere G4 (PDB code: 2gku). The figures were adapted with permission from Reference 56. PDB; Protein Data Bank.

**Figure 2 molecules-24-00429-f002:**
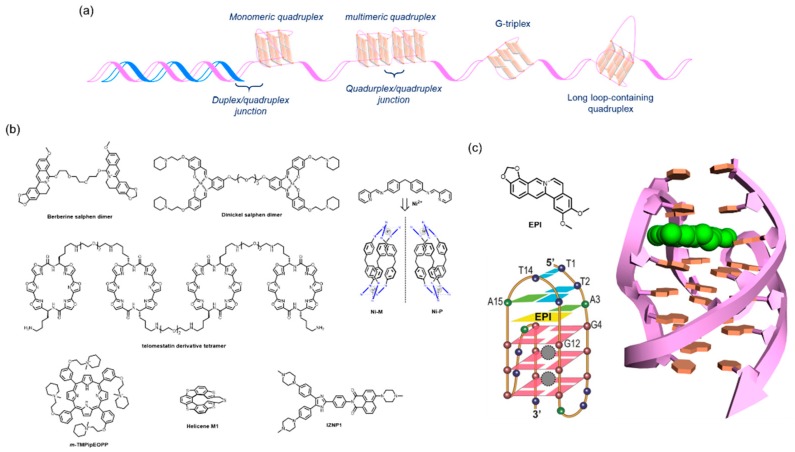
Selective telomere G4 targeting by ligands. (**a**) Telomere G-stretch sequences potentially adopt non-canonical G4s that offer specific binding motifs. (**b**) Several telomere G4-preferred binders based on the specific-motif recognition. It is worth noting that the by Ni-M and IZNP1 exhibit differential antitumor activities, likely based on the specific-motif recognition of telomere G4s. (**c**) EPI and the solution structure of the EPI-wtTel26 complex (PDB code: 6ccw). The bases that do not participate in the interaction event with EPI are omitted for clarity. Figure 2a,c were adapted with permission from References 56 and 86, respectively.

**Figure 3 molecules-24-00429-f003:**
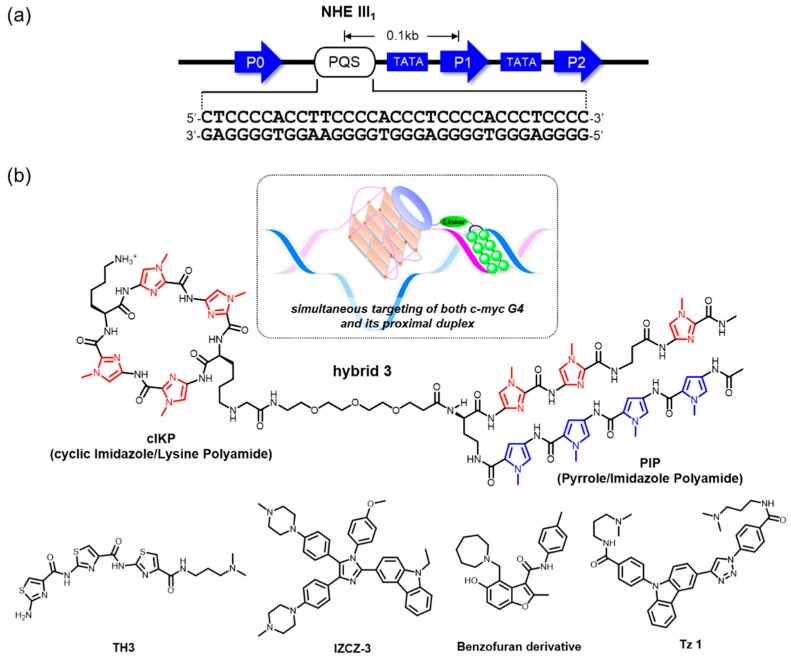
(**a**) The c-*MYC* promoter has one putative G4-forming sequence (PQS). (**b**) Molecules preferentially interacting with its G4 DNAs over other ones. *N*-Methylpyrrole is highlighted in blue and *N*-methylimidazole is highlighted in red. Antitumor activities of TH3, IZCZ-3, benzofuran derivative, and Tz 1 are summarized in Table 1.

**Figure 4 molecules-24-00429-f004:**
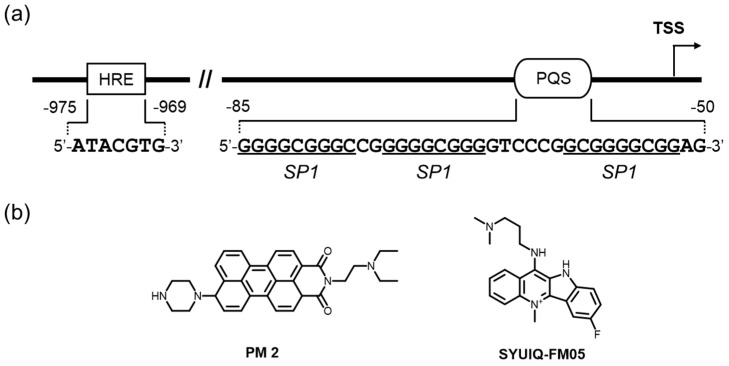
(**a**) The *VEGF* promoter has one putative G4-forming sequence (PQS) located close to the transcription start site (TSS) and hormone response element (HRE) that regulate the transcription. (**b**) Molecules interacting with its G4 DNAs and efficiently suppressing VEGF protein expression. It is worth noting that SYUIQ-FM05 has potential to reduce VEGF-stimulated tumor angiogenesis.

**Figure 5 molecules-24-00429-f005:**
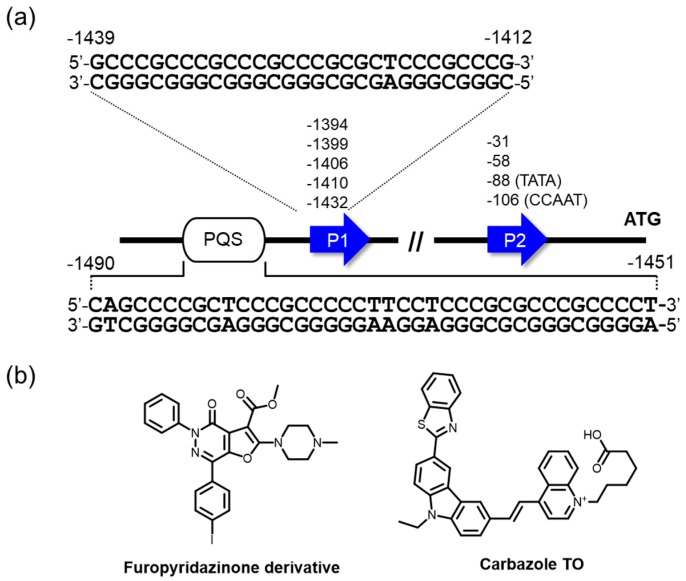
(**a**) The BCL2 promoter has two G4-forming elements that were shown to attenuate the BCL2 promoter activity. (**b**) Molecules preferentially interacting with the G4 (distal one) over other G4 structures. An antitumor activity of furopyridazinone derivative is summarized in Table 1.

**Figure 6 molecules-24-00429-f006:**
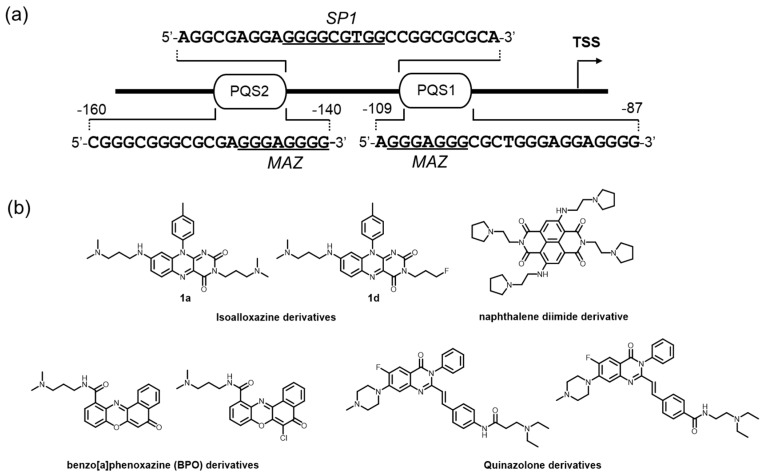
(**a**) The c-*kit* promoter has two PQSs, where several transcription factors are likely involved. (**b**) Molecules interacting with its G4 DNA and showing the downregulation of the c-*kit* gene transcription.

**Figure 7 molecules-24-00429-f007:**
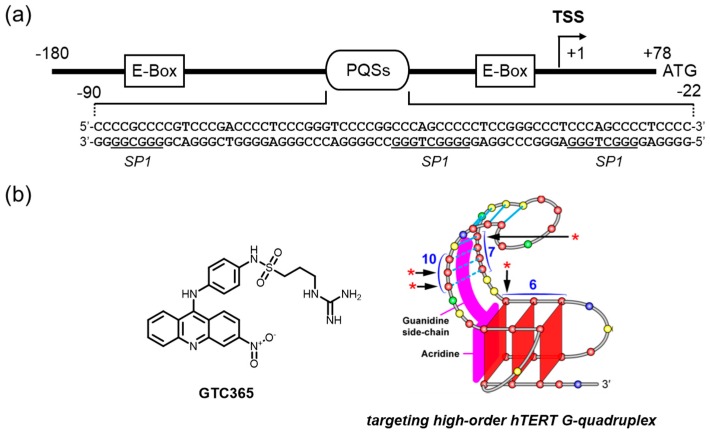
(**a**) The *hTERT* promoter has several PQSs, where two tandemly aligned G4s are suggested, and a molecule interacting with the specific motif of its G4 DNA and showing the downregulation of the *hTERT* gene transcription. The arrows with asterisks in the right illustration of the (**b**) represent bases that are protected in the presence of GTC365 from methylation by dimethyl sulfate (DMS) in the experiment performed in Reference 102. The protection suggests the occupied site of GTC365. The numbers represent the order of the runs of poly G observed in the hTERT promoter, which are numbered in Reference 102. The antitumor activity of GTC365 is summarized in Table 1. The right illustration of the Figure 7 (**b**) was adapted with permission from Reference. 172.

**Figure 8 molecules-24-00429-f008:**
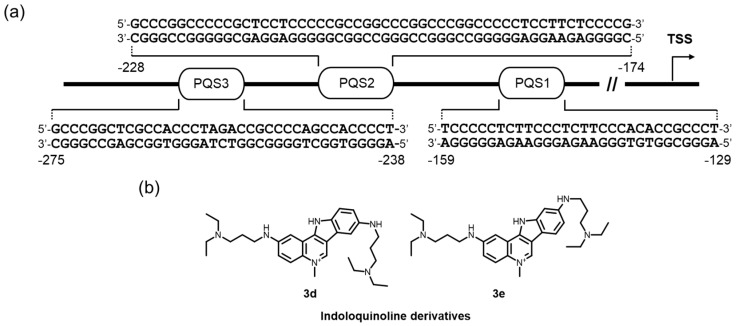
(**a**) The human KRAS promoter has three PQSs. In particular, the G4 formation at the most proximal PQS (PQS1) is shown to act as a stronger transcriptional repressor. (**b**) Molecules interacting with its G4 DNA and causing antitumor activities.

**Figure 9 molecules-24-00429-f009:**
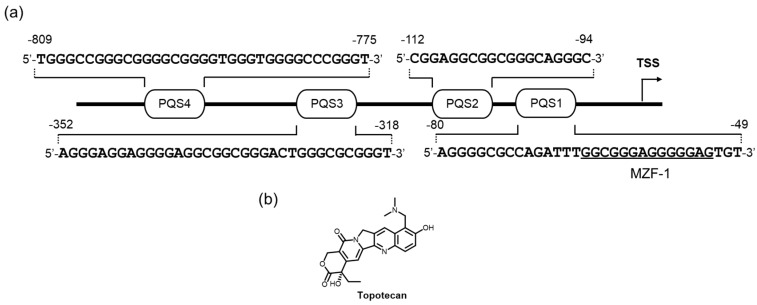
(**a**) The c-*myb* promoter has multiple PQSs. (**b**) A molecule interacting with its G4 DNA and efficiently repressing MYB protein expression.

**Figure 10 molecules-24-00429-f010:**
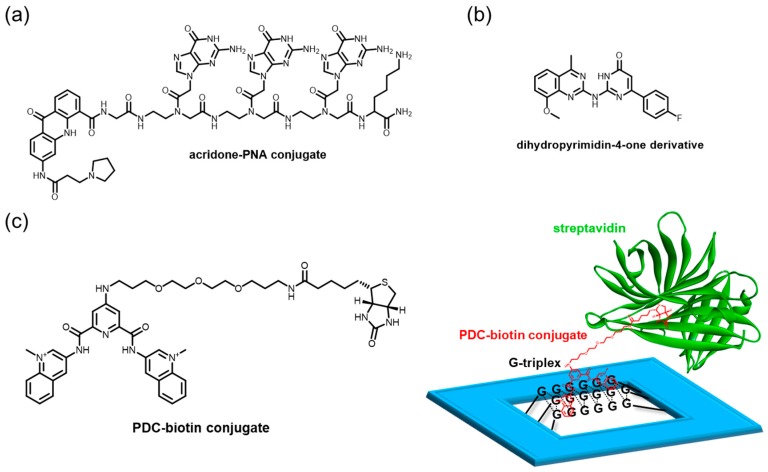
(**a**–**c**) G-triplex-targeting ligands. (**c**) A platform for their evaluation constructed by DNA origami.

**Figure 11 molecules-24-00429-f011:**
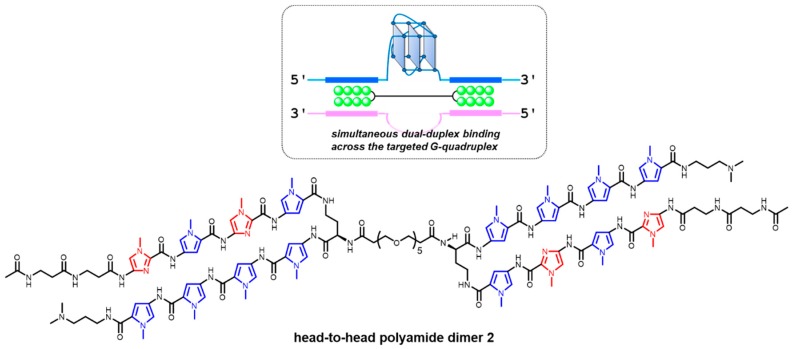
Head-to-head polyamide dimer that can target a certain G4 by simultaneously binding to dual-duplex sites across the G4. *N*-Methylpyrrole is highlighted in blue, and *N*-methylimidazole is highlighted in red.

**Table 1 molecules-24-00429-t001:** Summary of targeted G-quadruplex (G4)-preferred ligands that exhibit antitumor activities.

Ligand ^a^	Target G4 or Gene ^b^	Preferred Target Structure ^c^	Cell Line ^d^	Tumor Type	Effect	Notes	References
Ni-P	telomere	hybrid	MDA-MB-231	breast cancer (adenocarcinoma)	· Caner stem cell-specific apoptosis · Bulk cancer-specific apoptosis and senescence · Negligible cytotoxicity to normal somatic cells	· Tumor growth suppression in a MDA-MB-231 xenograft model vivo	[82,83]
			MCF-7	breast cancer (adenocarcinoma)			
IZNP1	telomere	dimeric G4s	SiHa	squamous cell carcinoma	· Apoptosis · Senescence	·Telomere dysfunction (DNA damage and telomere uncapping)	[85]
TH3	c-*MYC*	c-*MYC* (parallel)	A549	lung cancer	· Antiproliferative effect (apoptosis) · Negligible cytotoxicity to normal somatic cells	· Validation of the minimal effects for a G4-driven gene (BCL2)	[97]
			Hela	cervical cancer			
IZCZ-3	c-*MYC*	c-*MYC* (parallel)	SiHa	squamous cell carcinoma	· Antiproliferative effect (apoptosis) · Negligible cytotoxicity to normal somatic cells	· Validation of c-*MYC* G4-dependent gene suppression · Tumor growth suppression in a SiHa xenograft model in vivo	[98]
			Hela	cervical cancer			
			Huh7	liver cancer			
			A375	malignant melanoma			
Benzofuran derivative	c-*MYC*	c-*MYC* (parallel)	L363 MM1S MM1R etc.	myeloma	· Antiproliferative effect (apoptosis) · Negligible cytotoxicity to normal cells	· Validation of the minimal effects for other G4-driven genes	[102]
Tz 1	c-*MYC*	c-*MYC* (parallel)	HCT116	colorectal carcinoma	· Apoptosis	· Validation of c-*MYC* G4-dependent gene suppression	[103]
Furopyridazinone derivative	BCL2	BCL2 (hybrid)	Jurkat	human acute T cell leukemia	· Antiproliferative effect (apoptosis) · Negligible cytotoxicity to normal cells	-	[135]
GTC365	hTERT	hTERT (stem-loop-containing hybrid)	MCF-7	breast cancer (adenocarcinoma)	· Apoptosis · Senescence	· Validation of decreased telomerase activity and telomere length	[172]

^a^ This column describes G-quadruplex (G4)-interacting ligands that were reported to exhibit antitumor activities likely based on the clear selectivity to target G4s. ^b^ This column a G4-related genomic structure (telomere) or G4-driven genes the were intended to be targeted by each ligand. ^c^ This column describes G4 structures or topologies that were preferred by each ligand among other G4s examined in the respective papers. ^d^ This column describes cell lines in which antitumor activities of ligands were examined.

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
