# Peer review of "Recent Progress of Targeted G-Quadruplex-Preferred Ligands Toward Cancer Therapy"

_molecules, 2019, doi:10.3390/molecules24030429_

Round 1
Reviewer 1 Report
Dear Authors,
Please find below the Reviewer`s comments.
l.10 The abstract is too short. Add 2-3 sentences more.
l.244 expand the abbreviations when appear for the first time.
l.400 move the figures to the correct positions.
I suggest to create the table where the following information will be placed: selective G4-targeting ligands (all or the most important), gene target, cellular/organ/organism effects.
l. 530-537 move the paragraph to the introduction.
The reviewer
Author Response
Dear Dr. Li,
Molecules
We would like to re-submit a revised version of our manuscript titled Recent progress of specific G-quadruplex-preferred ligands toward cancer therapy (Manuscript ID: molecules-410205)to this journal.
Based on the reviewers’ comments, we have revised the original text while at the same time maintaining the high standard of the scientific content. We have commented on the modified parts in the revised manuscript. We hope that our revised version is now suitable for publication. The point-by-point responses are described below.
Reviewer 1:
1. l.10 The abstract is too short. Add 2-3 sentences more.
l.244 expand the abbreviations when appear for the first time.
l.400 move the figures to the correct positions.
Ø All minor critiques have been reflected in the revised manuscript.
2. I suggest to create the table where the following information will be placed: selective G4-targeting ligands (all or the most important), gene target, cellular/organ/organism effects.
Ø We thank the reviewer for his/her suggestion. We have created the summary table (Table 1) of specific G4-preferred ligands that exhibit antitumor activities.
3. l. 530-537 move the paragraph to the introduction.
Ø We have added the content of this paragraph to the introduction with a small modification;
Relatively recent studies revealed that G4 also had an impact on differentiation- and neuron-related genes [20]. For instance, OCT4 expression may be governed to some degree by G4 formation at the proximal promoter in human embryonic stem cells (CCTL14) [20a], whereas the excessive formation of repetitive G-quadruplex structures on an expandable (GGGGCC)nin C9orf72gene or (CGG)nin FMR1gene accounts for some neurogenetic disorders [20b]. On the contrary, G4 can act positively in neurons, whereG4 structures at the CpG island located in xl3b are recognized by ATRX, contributing to appropriate synaptic function [20c].
Reviewer 2 Report
More than 370000 sequences have been recognized to form potential G4 structure in the human genomes, especially in G-rich regions of telomeres, promoter regions, introns, immunoglobulin switch regions, and more. G4 structures are mostly involved in regulating various biological processes such as maintenance of genetic information, replication, transcription, cell aging, translation, and diseases such as cancer. Thus, G-quadruplexes are considered as an important target for the development of anti-cancer drugs and therapeutic applications. Moreover, down-regulation of transcription has been observed with small molecules that are able to induce and stabilize G4 structures in the promoters of several oncogenes. Therefore, several G4 binding small molecules have been developed during the last two decades. Many of these ligands are selective for G4 structures over duplex DNA, but the design of a ligand specific for a given G4 structure is still challenging.
In the submitted paper authors reviewed recent developments in the field of G-quadruplex ligands. They discussed the selectivity of ligands for cancer-related sequences and the review is focused mainly on a biomedical issues. The content is interesting and timely but readers with chemistry background involved in drug design and development would expect more information concerning topology of targeted G4 structures, critical comments on binding sites and conclusions related to parameters that decide on the specific binding of ligand to a particular G4 structure. Such data are missing or are scarce. An additional paragraph that summarizes miscellaneous methods and techniques exploited for G4 ligands studies would be appreciated.
Other points:
1. Ref. [82] – I would suggest to replace this ref with other paper of these authors (J. Am. Chem. Soc., 2014, 136 (11), pp 4161–4171)
2. Figures need more details in captions:
- Fig. 4: Ref. XX?
- Fig. 7: caption is before the figure, “molecule” not “molecules”, correct “tageting” to “targeting” , the complex hTERT/ligand GTC365 shown in figure needs more explanation, e.g., meaning of arrows, asterisks and numbers
- Fig. 9: caption is before the figure
Figures 10 and 11 overlapped.
Author Response
Please find the attached manuscript.

Reviewer 3 Report
This revised review article "Recent progress of specific G-quadruplex-preferred ligands towards cancer therapy" provides a summary of select works done across an array of G4-containing/forming DNA regions, as they relate to cancer. Overall, however, the review lacks many relevant and important references, lacks sufficient detail to be understandable to a broad audience and lacks depth in content. There have been many, many reviews on G4-targeted therapies and the current manuscript does not particularly add to the field.
In particular, the references lacking include the body of works by Shankar Balasubramanian's group demonstrating G4-formation in RNA and DNA, in telomeres, centromeres, and promoters, and as induced by Pyridostatin in cells and even in patient biopsies. Shankar's group's work regarding pulling down G4 sequences from cells and showing the array of consensus and non-consensus sequences is also relevant to page 1, line 29. Page 2, line 55 should reference the review by Hurley, Neidle and Balasubriamanian in the discussion of quorfloxin, which should be updated to reflect the end of phase 2 studies over 5 years ago. There is a lack of discussion of the G4-interactive compound currently in clinical trials, CX5461. The MYC section requires a discussion of the exon test development and the ellipticine described in that work by Brown et al in 2011, and the non-G4 targeted development by Hurley in Bodupally, 2012. The description of supercoiling in the MYC promoter needs a discussion of the work by Sun et al in 2009, and the contribution of the Sun group is also lacking in the VEGF promoter description. The Bcl-2 promoter contains more than just the one G4 described, including one at the P1 promoter described by Yang's group, and c-KIT has two promoter G4s as described by Neidle's group, that are not included. The hTERT promoter has two G4s, described by Palumbo, and the description of the multiple kRAS promoter G4s was done by the Brooks group in 2016, which is also missing.
In reference to the lack of detail and depth, the telomeric description discusses mono-, di-, and multi-meric structures with no description and misconstrues compounds as specific when they are, at best, selective, lacks a complete description of in cell/biological activities. The VEGF, Bcl2, hTERT, and c-myb sections contain a cursory explanation of any G4s and selective compounds, and the "others" section lacks many other promoters, including those relevant to the review, such as VEGFR2 (KDR). This manuscript is missing too much discussion of the G4 field for a person familiar with the field, and not enough description for a person outside the field.
Other concerns include an out of place discussion of neurologically relevant G4s (in a cancer focused manuscript), a lack of inclusion of RNA G4s, and an out of place discussion of triplexes. Moreover, there are oligonucleotides targeting G4s that aren't discussed completely, although a few PNAs are mentioned.
As a minor point, the manuscript needs moderate editing for word choice in English, as well as plural agreement.
Author Response
Reviewer 3:
1. This revised review article "Recent progress of specific G-quadruplex-preferred ligands
towards cancer therapy" provides a summary of select works done across an array of G4-
containing/forming DNA regions, as they relate to cancer. Overall, however, the review lacks
many relevant and important references, lacks sufficient detail to be understandable to a
broad audience and lacks depth in content. There have been many, many reviews on G4-
targeted therapies and the current manuscript does not particularly add to the field.
Ø We thank the reviewer for his/her comment. As mentioned in the abstract and introduction
section, our manuscript intends to review G4 DNA-ligands exhibiting a considerable
anticancer effect that is likely explained by a clear binding preference to certain G4s. To the
best of our knowledge, there have never been reports regarding this, except for our recent
review article that mainly focuses on selective G4 ligands among different G4s and unique
ligand design methodology (Chem Eur J 2018 in press). Revision of our manuscript here has
been made to more clearly reflect these concepts and to consider the reviewer 3’s comments.
2. In particular, the references lacking include the body of works by Shankar Balasubramanian's
group demonstrating G4-formation in RNA and DNA, in telomeres, centromeres, and
promoters, and as induced by Pyridostatin in cells and even in patient biopsies. Shankar's
group's work regarding pulling down G4 sequences from cells and showing the array of
consensus and non-consensus sequences is also relevant to page 1, line 29
Ø According to his/her suggestions, we have additionally added to the revised manuscript
papers by prof. Sir Shankar Balasubramanian’s group that address the evidence of G4s in
cells using small ligands or G4 antibodies (ref. 7e and 21a).
3. Page 2, line 55 should reference the review by Hurley, Neidle and Balasubriamanian in the
discussion of quorfloxin, which should be updated to reflect the end of phase 2 studies over 5
years ago.
Ø Done. See the revised text (page 2, line 56-57).
4. There is a lack of discussion of the G4-interactive compound currently in clinical trials,
CX5461.
Ø Done. See the revised text (page 2, line 78-79).
5. The MYC section requires a discussion of the exon test development and the ellipticine
described in that work by Brown et al in 2011, and the non-G4 targeted development by
Hurley in Bodupally, 2012. The description of supercoiling in the MYC promoter needs a
discussion of the work by Sun et al in 2009, and the contribution of the Sun group is also
lacking in the VEGF promoter description. The Bcl-2 promoter contains more than just the
one G4 described, including one at the P1 promoter described by Yang's group, and c-KIT
has two promoter G4s as described by Neidle's group, that are not included. The hTERT
promoter has two G4s, described by Palumbo, and the description of the multiple kRAS
promoter G4s was done by the Brooks group in 2016, which is also missing.
Ø We have now added to the revised manuscript references of the exon test assays developed
by Brooks and Hurley groups, which allows for assessing whether a G4 ligand directly targets
a c-myc G4 and downregulates the gene or not. Also, the VEGF, BCL2, c-KIT, hTERT, and
KRAS sections of the revised manuscript have been enriched according to his/her
suggestions. See the revised text.
6. In reference to the lack of detail and depth, the telomeric description discusses mono-, di-,
and multi-meric structures with no description and misconstrues compounds as specific when
they are, at best, selective, lacks a complete description of in cell/biological activities. The
VEGF, Bcl2, hTERT, and c-myb sections contain a cursory explanation of any G4s and
selective compounds, and the "others" section lacks many other promoters, including those
relevant to the review, such as VEGFR2 (KDR).
Ø We have now added a description about the selectivity of a ligand (topotecan) for clarity. See
the revised text (page 14, line 490-491).
7. Other concerns include an out of place discussion of neurologically relevant G4s (in a cancer
focused manuscript), a lack of inclusion of RNA G4s, and an out of place discussion of
triplexes. Moreover, there are oligonucleotides targeting G4s that aren't discussed completely,
although a few PNAs are mentioned.
Ø This manuscript omitted discussions of the RNA G4s to avoid the manuscript to be
complicated.
We had just added the description about neurologically relevant G4s to the introduction and
the last section with the intention of mentioning the versatility of G4s in cellular events.
Regarding the triplexes, we made this section because we believe that such structures are
amenable to specific recognition by ligands and will be exploited for targets toward specific
antitumor therapy.
Since, as the reviewer mentioned, oligonucleotides targeting G4 DNAs are also attractive,
we mentioned it in KRAS section of the original text on the basis of the concept of this
manuscript. PNAs mentioned in the original text was selected on the basis of the concept of
this manuscript.

Round 2
Reviewer 3 Report
The revised version of the review manuscript, "Recent Progress of specific G-quadruplex-preferred ligands toward cancer therapy" is markedly expanded since the first review. There remain a few items that require address, but overall it is more prepared for publication.
Items that require address:
Global items persist in acronym and name representation choices. For example, the authors toggle between "G-quadruplex" and "G4" and between "c-myc", "Myc" and "MYC". Editing for consistency is suggested.
Globally, the authors are advised to change the majority of their use of the word "specific" to "selective" and "highly selective". There are no compounds specific for one G4 over another or truly for G4s over all other DNA topologies. The BG4 antibody is perhaps the only exception, but it is not discussed. This includes a suggestion to amend the title.
The addition of a sentence for Quofloxin is not appropriate. The authors should state the findings of the Phase II trials that were halted due to high albumin binding. This information is available online and can be referred to.
The discussion of Bcl-2 and the G4s is lacking significant findings, particularly that of the Yang group with the 2016 JACS paper by Onel describing another, discrete, G4 at the P1 promoter. Other works missing in discussion include the 2011 Brown et all paper describing the initial CA46 exon test for the compound GQC-05 (NSC338258). Those finding are pertinent to the c-myc section. Missing references are also present in the kRAS section (namely the 2016 paper by Morgan being the first to describe other promoter G4s). The hTERT section is missing an entire discussion of any G4 formation (it jumps right into the compounds), and particularly the description of two distinct tandem G4s. The section on c-myb misleadingly (line 475) suggests that the GC-rich promoter is only present in bone and blood cells, versus in all cells in the body. Line 486 requires an edit for "MDF7" cells to read "MCF-7" cells.
Author Response
Based on Reviewer 3’s comments, we have revised the text while at the same time maintaining the high standard of the scientific content. We have commented on the modified parts in the revised manuscript. We hope that our revised version is now suitable for publication. The point-by-point responses are described below.
Reviewer 3:
1. Global items persist in acronym and name representation choices. For example, the authors toggle between "G-quadruplex" and "G4" and between "c-myc", "Myc" and "MYC". Editing for consistency is suggested.
Ø We thank the reviewer for his/her comment. We have edited the text for consistency, including G-quadruple/G4 (adjusted to G4, except for the title and the subtitles), c-myc/myc/MYC (adjusted to c-MYC).
2. Globally, the authors are advised to change the majority of their use of the word "specific" to "selective" and "highly selective". There are no compounds specific for one G4 over another or truly for G4s over all other DNA topologies. The BG4 antibody is perhaps the only exception, but it is not discussed. This includes a suggestion to amend the title.
Ø We thank the reviewer for his/her comment. We have carefully checked the words, “specific”, “specificity”, and “specifically” throughout the text, and reworded them to “selective”, “selectivity”, “selectively”, and “preferentially” (or omitted in some cases) when these words are used in the context of the actual binding property of the ligands.
3. The addition of a sentence for Quofloxin is not appropriate. The authors should state the findings of the Phase II trials that were halted due to high albumin binding. This information is available online and can be referred to.
Ø We thank the reviewer for letting us know the reason for not proceeding to the Phase III trials of Qualfloxin after the completion of the Phase II trials, since we were not able to get the information about this. We have then found a book that includes an important description of such things and now cited it in the revised manuscript. (R. K. Morgan and T. A. Brooks Targeting Promoter G-Quadruplexes for Transcriptional Control (Chapter 7) in Small-molecule Transcription Factor Inhibitors in Oncology, 2016).
4. The discussion of Bcl-2 and the G4s is lacking significant findings, particularly that of the Yang group with the 2016 JACS paper by Onel describing another, discrete, G4 at the P1 promoter. Other works missing in discussion include the 2011 Brown et all paper describing the initial CA46 exon test for the compound GQC-05 (NSC338258). Those finding are pertinent to the c-myc section. Missing references are also present in the kRAS section (namely the 2016 paper by Morgan being the first to describe other promoter G4s). The hTERT section is missing an entire discussion of any G4 formation (it jumps right into the compounds), and particularly the description of two distinct tandem G4s. The section on c-myb misleadingly (line 475) suggests that the GC-rich promoter is only present in bone and blood cells, versus in all cells in the body. Line 486 requires an edit for "MDF7" cells to read "MCF-7" cells.
Ø
We thank the reviewer for his/her suggestions and comments. We have now added a new G4-forming sequence immediately upstream of the P1 promoter to Figure 5a as depicted below:
Also, we have clearly mentioned the new BCL2 G4 and its relevance to the promoter activity in the revised manuscript as follows;
More recently, Onel, Yang, and coworkers demonstrated by a luciferase reporter assay using BCL2 promoter and the mutated sequences that the formation of another G4 situated almost on the upper region of P1 promoter attenuated the promoter activity (Figure 5a) [76b].
Regarding the c-MYC part, we have created a paragraph for the exon-specific assays devised by Brooks group and GQC-05 in the revised manuscript as follows;
An exon-specific RT-qPCR assay using two pairs of Burkitt’s lymphoma cell lines (CA46 and RAJI) devised by Brooks group allowed us to evaluate whether G4 ligands directly act for c-MYC G4 or not in cells [49]. This assay relies on the unique reciprocal translocation of c-MYC gene locus in the two cell lines; by the translocation in CA46 where an exon 1 is separated from exons 2 and 3, the G4-driven transcriptional activity was only maintained for exon 1, whereas these three exons and the PQS remains tandemly after the translocation in RAJI. That is why the use of primers specific to the exons 1 and 2 in a qPCR permits the demonstration of the c-MYC G4-mediated control by G4 ligands in the cellular context. In fact, a G4-interacting ligand, GQC-05 (NSC338258) was shown to directly suppress the gene expression in a c-MYC G4-mediated manner using this assay [49a]. This system is now a standard method for studying the intracellular activities of the c-MYC G4 targeting small molecules.
Regarding the KRAS part, we have added descriptions of newly studied G4s in the KRAS promoter and their influence on the promoter activity as follows;
More recently, comprehensive studies on the human KRAS promoter in terms of the possible G4 structures and their relevance to the promoter activity that were performed by Brooks group revealed that a newly discovered G4 formed in the PQS2 more critically affected the attenuation of the promoter (Figure 8a) [17b]. These findings would contribute to the creation of more efficient and selective G4-mediated transcriptional repression by ligands in a future study.
Regarding the hTERT part, we have added descriptions of the G4 formations in the revised manuscript as follows;
The region located -90 ~ -22 upstream of the transcription start site was abundant in GC base pairs and has multiple PQSs (Figure 7a) [15]. In particular, tandemly aligned G4s formed by the entire PQSs have been suggested to be a key mechanism to maintain the normal transcriptional levels of the hTERT gene [15,102]. Interestingly, the involvement of the dynamic equilibrium of such three-dimensional structures of DNA upon the somatic mutations (G/C to T/A) in this region is attributable to activated TERT expression [102].
Regarding the c-myb part, we have now deleted a word “in bone and blood cells” for clarity and replaced all “MDF7” with “MCF-7” in the revised manuscript.
